# A simple tensor network algorithm for two-dimensional steady states

Augustine Kshetrimayum[1], Hendrik Weimer[2] & Román Orús[1]

Understanding dissipation in 2D quantum many-body systems is an open challenge which has proven remarkably difficult. Here we show how numerical simulations for this problem are possible by means of a tensor network algorithm that approximates steady states of 2D quantum lattice dissipative systems in the thermodynamic limit. Our method is based on the intuition that strong dissipation kills quantum entanglement before it gets too large to handle. We test its validity by simulating a dissipative quantum Ising model, relevant for dissipative systems of interacting Rydberg atoms, and benchmark our simulations with a variational algorithm based on product and correlated states. Our results support the existence of a first order transition in this model, with no bistable region. We also simulate a dissipative spin 1/2 *XYZ* model, showing that there is no re-entrance of the ferromagnetic phase. Our method enables the computation of steady states in 2D quantum lattice systems.

[1] Institute of Physics, Johannes Gutenberg University, 55099 Mainz, Germany. [2] Institut für Theoretische Physik, Leibniz Universität Hannover, Appelstr. 2, 30167 Hannover, Germany. Correspondence and requests for materials should be addressed to R.O. (email: roman.orus@uni-mainz.de)

Understanding the effects of dissipation in quantum many-body systems is an open challenge. When the quantum system is immersed in an environment and coupled to it, the exchange of information (e.g., energy, heat, and particles) between system and environment usually leads to dissipation when the environment is larger than the system. If the dissipation is Markovian (i.e., if no information flows back into the system), then the evolution is generated by a Liouvillian superoperator $\mathcal{L}$, and can be casted in the form of a master equation for the reduced density matrix of the quantum system. As time flows, the system dissipates, until reaching in many cases a steady, or "dark" state $\rho_S$, so that $\mathcal{L}[\rho_s] = 0$. This process is important in several contexts, e.g., understanding the decoherence of complex wavefunctions[1], quantum thermodynamics[2], engineering of topological order through dissipation[3], and driven-dissipative universal quantum computation[4]. The study of non-equilibrium quantum complex systems has recently received much attention[5–9].

In this paper we present a method to approximate such steady states for 2D quantum lattice systems of infinite size (i.e., in the thermodynamic limit). Over the years, the solution to this problem has proven remarkably difficult. Our method is to be compared to alternatives in 2D such as cluster mean-field methods[10], correlated and product state variational ansatzs[11,12], and corner space renormalization group[13]. Importantly, none of these methods targets the truly 2D quantum correlations that are present in the problem. The method that we propose here is based on tensor networks (TN)[14–18] and is, in fact, particularly simple and efficient. Whereas TN methods have been used in the context of dissipative 1D systems[19–21] and thermal 2D states[22,23], our method uses truly 2D TNs to target 2D dissipation. To prove the validity of our algorithm, we compute the steady states of the dissipative 2D quantum Ising model for spin 1/2, which is of relevance for controversies concerning dissipation for interacting Rydberg atoms[11]. As we shall discuss, we compare our results with those obtained by a variational algorithm based on product and correlated states[12]. Moreover, we also simulate a dissipative spin 1/2 *XYZ* model, showing that there is no re-entrance of the ferromegnatic phase, compatible with recent cluster mean-field results[10].

## Results

### Parallelism with imaginary time evolution

We start by considering a master equation of the form

$$\dot{\rho} = \mathcal{L}[\rho] = -i[H, \rho] + \sum_\mu \left( L_\mu \rho L_\mu^\dagger - \frac{1}{2}\left\{ L_\mu^\dagger L_\mu, \rho \right\} \right), \quad (1)$$

where $\rho$ is the density matrix of the system, $\mathcal{L}$ is the Liouvillian superoperator, $H$ the Hamiltonian of the system, and $\{L_\mu, L_\mu^\dagger\}$ the Lindblad operators responsible for the dissipation. Following a similar approach as in Zwolak and Vidal[24], we can also write the same equation in vectorized form using the so-called "Choi's isomorphism", i.e., understanding the coefficients of $\rho$ as those of a vector $|\rho\rangle_\#$ (intuitively, $|a\rangle\langle b| \simeq |a\rangle \otimes |b\rangle$, Fig. 1a): $|\dot{\rho}\rangle_\# = \mathcal{L}_\# |\rho\rangle_\#$, where the "vectorized" Liouvillian is given by

$$\mathcal{L}_\# \equiv -i(H \otimes \mathbb{I} - \mathbb{I} \otimes H^T)$$
$$+ \sum_\mu \left( L_\mu \otimes L_\mu^* - \frac{1}{2} L_\mu^\dagger L_\mu \otimes \mathbb{I} - \frac{1}{2} \mathbb{I} \otimes L_\mu^* L_\mu^T \right). \quad (2)$$

In the above equation, the symbol of tensor product $\otimes$ separates operators acting on either the l.h.s. (ket) or the r.h.s. (bra) of $\rho$ in its matrix form. Whenever $\mathcal{L}_\#$ is independent of time, time evolution can be formally written as $|\rho(T)\rangle_\# = e^{T\mathcal{L}_\#} |\rho(0)\rangle_\#$, which for very large times T may yield

a steady state $|\rho_s\rangle_\# \equiv \lim_{T\to\infty} |\rho(T)\rangle_\#$. It is easy to see that the state $|\rho_s\rangle_\#$ is the eigenvector of $\mathcal{L}$ corresponding to zero eigenvalue, so that $\mathcal{L}_\# |\rho_s\rangle_\# = 0$.

Next, let us consider the special but quite common case in which the Liouvillian $\mathcal{L}$ can be decomposed as a sum of local operators. For nearest-neighbor terms, one has the generic form $\mathcal{L}[\rho] = \sum_{\langle i,j\rangle} \mathcal{L}^{[i,j]}[\rho]$, where the sum $\langle i, j\rangle$ runs over nearest neighbors. In the "vectorized" notation (#), this means that $\mathcal{L}_\# = \sum_{\langle i,j\rangle} \mathcal{L}_\#^{[i,j]}$.

The combination of the expressions above yields a parallelism with the calculation of ground states of local Hamiltonians by imaginary time evolution, which we detail in Table 1.

**Computing 2D steady states.** Given the parallelism above, it is clear that one can adapt, at least in principle, the methods to compute imaginary time evolution of a pure state as generated by local Hamiltonians, to compute also the real-time evolution of a mixed state as generated by local Liouvillians. This was, in fact, the approach taken by Zwolak and Vidal[24] for finite-size 1D systems, using Matrix Product Operators (MPO)[25] to describe the 1D reduced density matrix, and proceeding as in the Time-Evolving Block Decimation (TEBD) algorithm for ground states of 1D local Hamiltonians[26,27].

Inspired by the above, our method for 2D systems proceeds by representing the reduced density operator $\rho$ by a Projected Entangled Pair Operator (PEPO)[14–18] with physical dimension $d$ and bond dimension $D$, see Fig. 1b. Such a construction does not

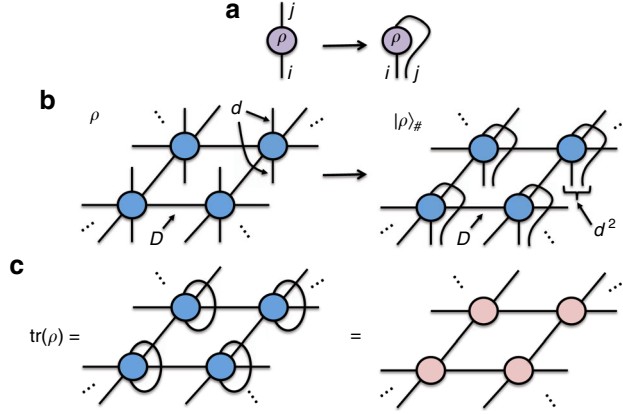

**Fig. 1** Relevant tensor network diagrams. **a** Tensor network diagram for the reduced density matrix $\rho$, with matrix elements $\rho_i^j$. The vectorization is, simply, reshaping the two indices into a single one; **b** tensor network diagram for the PEPO of $\rho$ on a 2D square lattice, with bond dimension $D$ and physical dimension $d$. When vectorized, it can be understood as a PEPS for $|\rho\rangle_\#$ with physical dimension $d^2$; **c** The trace of $\rho$ maps to the contraction of a 2D network of tensors

**Table 1 Parallelism between the calculation of ground states by imaginary time evolution, and the calculation of steady states by real-time evolution**

| Ground states | Steady states |
|---|---|
| $H = \sum_{\langle i,j\rangle} h^{[i,j]}$ | $\mathcal{L}_\# = \sum_{\langle i,j\rangle} \mathcal{L}_\#^{[i,j]}$ |
| $e^{-\tau H}$ | $e^{T\mathcal{L}_\#}$ |
| $|e_0\rangle$ | $|\rho_s\rangle_\#$ |
| $\langle e_0|H|e_0\rangle = e_0$ | $_\#\langle\rho_s|\mathcal{L}_\#|\rho_s\rangle_\# = 0$ |
| Imaginary-time $\tau$ | Real-time $T$ |

On the left hand side, $H$ is a Hamiltonian that decomposes as a sum of local terms $h^{[i,j]}$, $|e_0\rangle$ is the ground state of $H$ with eigenvalue $e_0$, and $\tau$ is the imaginary time

guarantee the positivity of the reduced density matrix[28]. However, we shall see later that this lack of exact positivity is not too problematic in our numerical simulations. Once vectorized, the PEPO can be understood as a Projected Entangled Pair State (PEPS)[29] of physical dimension $d^2$ and bond dimension $D$, as shown also in Fig. 1b. Next, we notice that for the case of an infinite-size 2D system, this setting is actually equivalent to that of the infinite-PEPS algorithm (iPEPS) to compute ground states of local Hamiltonians in 2D in the thermodynamic limit[30]. Thus, in principle, we can use the full machinery of iPEPS to tackle as well the problem of 2D dissipation and steady states.

There seems to be, however, one problem with this idea: unlike in imaginary time evolution, we are now dealing with real-time. In the master equation, part of the evolution is generated by a Hamiltonian $H$, and part by the Lindblad operators $L_\mu$. The Hamiltonian part corresponds actually to a unitary "Schrödinger-like" evolution in real-time, which typically increases the "operator-entanglement" in $|\rho\rangle_\#$, up to a point where it may be too large to handle for a TN representation (e.g., 1D MPO or 2D PEPO) with a reasonable bond dimension. In 1D this is the reason why the simulations of master equations are only valid for a finite amount of time. In 2D, simple numerical experiments indicate that in a typical simulation the growth of entanglement is even faster than in 1D.

Luckily, this is not a dead-end: if the dissipation is strong compared to the rate of entanglement growth, then the evolution drives the system into the steady state before hitting a large-entanglement region. The main point of this paper is to show that this is indeed the case for 2D dissipative systems. Regarding settings where dissipation is not so strong, our algorithm is a good starting point to compute steady states in the strong-dissipation regime. The strength of the dissipation can then be lowered down adiabatically, and using as initial state the one pre-computed for slightly-stronger dissipation. In this way one may get rid of local minima and obtain good results also in the weak dissipation regime.

With this in mind, our algorithm just applies the iPEPS machinery to compute the time evolution in 2D with a local Liouvillian $\mathcal{L}$ and some initial state. For the examples shown in this paper, we use the so-called simple update scheme[31] for the time evolution of the PEPO, Corner Transfer Matrices (CTM)[32–39] for the calculation of observables (other approaches[40–49] would also be equally valid here), and random initial states. To check whether we have a good approximation of a steady state or not we compute the parameter $\Delta \equiv {}_\#\langle\rho_s|\mathcal{L}|\rho_s\rangle_\#$. For a good steady state approximation, this parameter should be close enough to zero, since we have $\Delta = 0$ in the exact case (in practice, we saw that the imaginary part of $\Delta$ is negligible, $\text{Im}(\Delta) \sim 10^{-15}$. Moreover, it should also be possible to check directly ${}_\#\langle\rho_s|\mathcal{L}^\dagger_\#\mathcal{L}_\#|\rho_s\rangle_\#$, but this is computationally more costly and does not change the conclusions of our observations). Another quantity that we used to check the validity of the simulations is the sum of negative eigenvalues of the (numerical) reduced density matrices of the system. More precisely, we define $\varepsilon_n \equiv \sum_{i|\nu_i<0} \nu_i(\rho_n)$, where $\rho_n$ is the reduced density matrix of $n$ contiguous spins in the steady state and $\nu_i(\rho_n)$ its eigenvalues, with only the negative ones entering the sum. In an exact case, this quantity should be equal to zero. However, the different approximations (operator-entanglement truncations) in the method may produce a small negative part in $\rho_s$, which can be easily quantified in this way (as a word of caution: notice that $\Delta$ and $\varepsilon_n$ can be used to benchmark our calculations, but they do not characterize the distance to the steady state. Moreover, in principle one could also develop a fully-positive algorithm for $\rho_n$[8], but at the expense of accuracy and efficiency[28]).

The computational cost of this algorithm is the one of the chosen iPEPS strategy. In our case, we work with a simple update for the evolution with a two-site unit cell, which has a cost of $O(d^4D^5+d^{12}D^3)$, and Trotter time steps $\delta t = 0.1$–$0.01$. The choice of Trotter steps actually depends on the time scales of the particular problem at hand. For the models considered here, we saw empirically that this choice was a good one. The convergence in the number of steps depends on the gap of the Liouvillian: the closer to a gapless point, the slower the convergence. Empirically we observed that this convergence was quite fast in the gapped phases of the models that we studied. Moreover, the CTM method for expectation values is essentially the one used to approximate classical partition functions on a 2D lattice (Fig.1c), which has a cost of $O(dD^4+\chi^2D^4+\chi^3D^3)$, being $\chi$ the CTM bond dimension. The overall approach is thus remarkably efficient. To have an idea of how efficient this is, let us imagine the following alternative strategy: we consider the Hermitian and positive semidefinite operator $\mathcal{L}^\dagger_\#\mathcal{L}_\#$, and target $|\rho\rangle_\#$ as its ground state. This ground state could be computed, e.g., by an imaginary time evolution. The problem, however, is that the crossed products in $\mathcal{L}^\dagger_\#\mathcal{L}_\#$ are non-local, and therefore the usual algorithms for time evolution are difficult to implement unless one introduces extra approximations in the range of the crossed terms[50]. Another option is to approximate the ground state variationally, e.g., via the Density Matrix Renormalization Group[51–54] or similar approaches[19,20] in 1D, or variational PEPS in 2D[29]. In the thermodynamic limit, however, this approach does not look very promising because of the non-locality of $\mathcal{L}^\dagger_\#\mathcal{L}_\#$ mentioned before. In any case, one could always represent this operator as a PEPO (in 2D), which would simplify some of the calculations, but at the cost of introducing a very large bond dimension in the representation of $\mathcal{L}^\dagger_\#\mathcal{L}_\#$. For instance, if a typical PEPO bond dimension for $\mathcal{L}_\#$ is ~4, then for $\mathcal{L}^\dagger_\#\mathcal{L}_\#$ it is ~16, which in 2D implies extremely slow calculations. Another option would be to target the variational minimization of the real part for the expectation value of $\mathcal{L}$[19,20]. This option, however, is also dangerous in 2D because of the presence of many local minima. In addition, the correct norm to perform all these optimizations is the one-norm of $\mathcal{L}(\rho)$ which, in contrast to the Òmore usualÓ 2-norm, is a hard figure of merit to optimize with variational TN methods. The use of real-time evolution is thus a safer choice in the context of the approximation of 2D steady states.

**Numerical simulations**. We first benchmark our method by simulating a dissipative spin 1/2 quantum Ising model on an infinite 2D square lattice, where dissipation pumps one of the spin states into the other. This model is of interest in the context of recent experiments with ultracold gases of Rydberg atoms[55,56]. Moreover, the phase diagram of its steady state is still a matter of controversy. Initially, it was predicted that the model exhibits a bistable phase[57,58], but several numerical and analytical calculations have cast doubts on this claim and predict instead a first order transition. In particular, a variational approach[11,12] and a Monte Carlo wavefunction approach[21] predict that the bistable phase is replaced by a first oder transition, which is also supported by arguments derived from a field-theoretical treatment of related models within the Keldysh formalism[59]. Furthermore, it is an open question whether the model supports an anti-ferromagnetic phase[11,12,57,60,61]. The master equation follows the one in Eq. (1), where the Hamiltonian part is given by $H = \frac{V}{4}\sum_{\langle i,j\rangle} \sigma_z^{[i]}\sigma_z^{[j]} + \frac{h_x}{2}\sum_i \sigma_x^{[i]} + \frac{h_z}{2}\sum_i \sigma_z^{[i]}$, with $\sigma_\alpha^{[i]}$ the $\alpha$-Pauli matrix at site $i$, $V$ the interaction strength, $h_x,h_z$ the transverse and parallel fields respectively, and where the sum over $\langle i,j\rangle$ runs over nearest neighbors. The dissipative part is given by operators $L_\mu = \sqrt{\gamma}\sigma_-^{[\mu]}$, so that in this particular case $\mu$ is a site index, and where $\sigma_-$ is the usual spin-lowering operator.

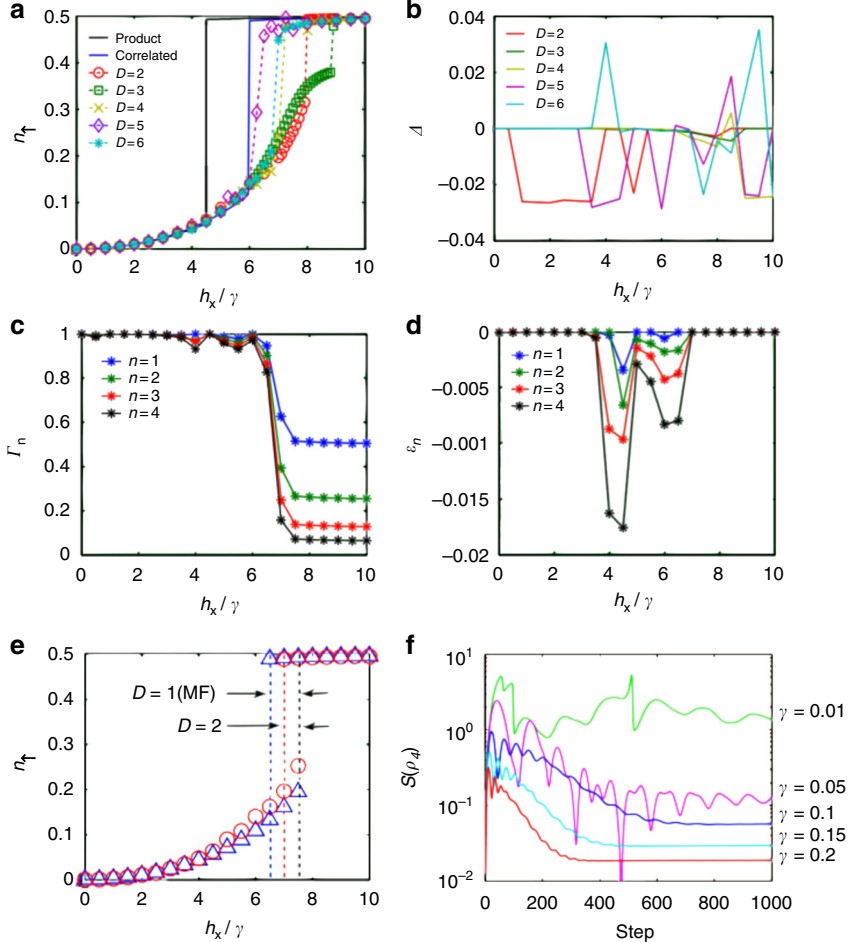

**Fig. 2** Computed quantities. **a** Spin-up density in the steady state as a function of $h_x/\gamma$ for $V = 5\gamma$, $\gamma = 0.1$ and $h_z = 0$, as computed with our method up to $D = 6$. For comparison, we show the results previously obtained by the variational method[12] with product states (black line) and correlated states (blue line); (**b**) $\Delta$ up to $D = 6$; (**c**) Purity $\Gamma_n$ of the reduced density matrix for a block of $n$ contiguous spins, for $D = 6$ (other bond dimensions have similar behavior). Spins are chosen within the $2 \times 2$ unit cell of the tensor network; (**d**) $\varepsilon_n$ of the reduced density matrix for a block of $n$ contiguous spins, for $D = 6$ (other bond dimensions have similar behavior). Overall, the convergence can be further improved by using more accurate update schemes; (**e**) bistable region for $D = 1$ (mean field) and $D = 2$. The region shrinks and disappears for larger bond dimension; (**f**) operator-entanglement entropy throughout the algorithmic evolution for a block a $2 \times 2$ unit cell with $D = 2$, $V = 0.5$, $h_x/\gamma = 10$, and different values of $\gamma$. The stronger the dissipation, the weaker the entanglement. A similar behavior is observed for larger D

In our simulations, we first set $V = 5\gamma$, $\gamma = 0.1$, $h_z = 0$ in order to compare with earlier results[12], which use a correlated variational ansatz with states of the form $\rho = \prod_i \rho_i + \sum_{\langle ij \rangle} C_{ij} \prod_{k \neq ij} \rho_k$, where $\rho_i$ are single site density matrices and $C_{ij}$ account for correlations. We compute the density of spins-up $n_\uparrow \equiv \sum_{i=1}^{N} \langle (1 + \sigma_z^{[i]}) \rangle / 2N$ ($N$ is the system's size) as a function of $h_x/\gamma$, for which it is believed to exist a 1st order transition in the steady state from a "lattice gas" to a "lattice liquid". This transition is clearly observed in our simulations in Fig. 2a, where simulations for $D = 5, 6$ agree with the correlated variational ansatz in the location of the transition point at $h_x^*/\gamma \sim 6$. In fact, as the bond dimension $D$ increases, we observe that there is more tendency towards agreeing with the correlated variational ansatz. We also observe a non-monotonic convergence in $D$, which may be due to a stronger effect of the approximations in the transition region, and which remains to be fully understood. Other quantities can also assess this transition, e.g., the purity of the $n$-site reduced density matrix $\Gamma_n \equiv \mathrm{tr}(\rho_n^2)$, which we plot in Fig. 2c for $D = 6$. We can see from that plot that the steady states $\rho_s$ for low $h_x/\gamma$ are quite close to a pure state (for which $\Gamma_n = 1 \forall n$). To validate this simulations we computed the

parameters $\Delta$ and $\varepsilon_n$ introduced previously, which we show in Fig. 2b and Fig. 2d respectively. One can see that $\Delta$ is always quite close to zero in our simulations, being at most $|\Delta| \sim 0.03$, so that the approximated $\rho_s$ is close to the exact steady state. Moreover, one can also see that $\varepsilon_n$ is always rather small, e.g., for $D = 6$ it is at most $\varepsilon_n \sim -0.017$ for the four-site density matrix close to the transition region (similar conclusions hold for other bond dimensions). This implies that the negative contribution to the numerical reduced density matrix is quite small, and therefore does not lead to large errors. In practice, we see that $\varepsilon_n$ seems to be extensive in $n$ away from the transition region, more specifically, $\varepsilon_n \sim n\varepsilon_0 + O(1/n)$, with $\varepsilon_0$ very close to zero. In our simulations we find a bistable region[12] for small $D$ that shrinks and disappears for $D > 2$, Fig. 2e, therefore being a unique steady state for large bond dimension. In Fig. 2f, we show the evolution of the four-site operator-entanglement entropy throughout the algorithm for increasing values of $\gamma$. The stronger the dissipation, the weaker the operator entropy (which never exceeds the support of the PEPO), and therefore the better the performance of the algorithm, as claimed.

Next, we introduce non-zero values of the parallel field $h_z$. In some regions of the phase diagram, mean-field and correlated state variational methods predict the existence of an "antiferromagnetic" (AF) phase, where $n_\uparrow$ attains different values between nearest neighbors in the square lattice[11]. In our

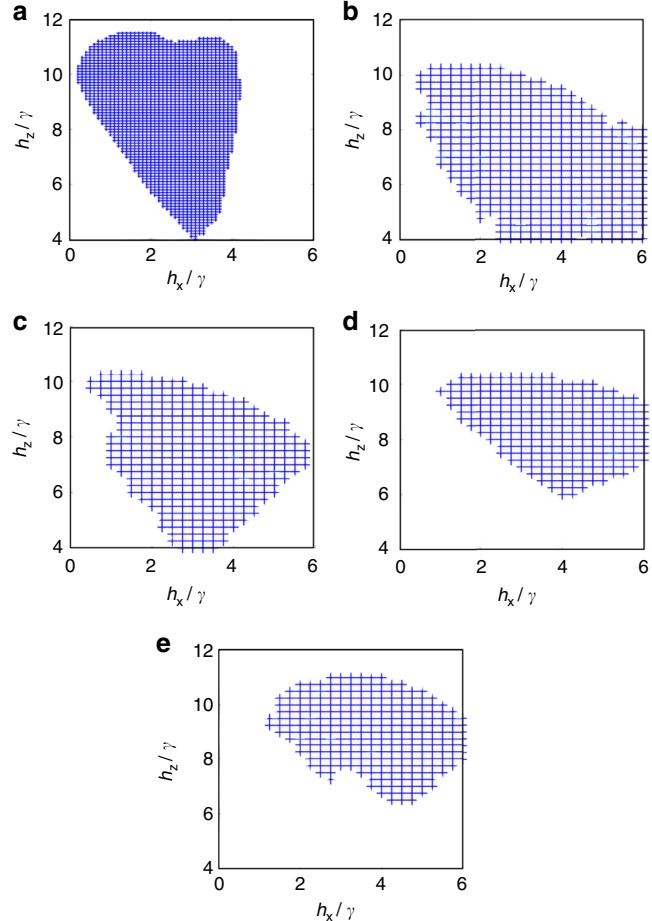

**Fig. 3** Antiferromagnetic region. In blue, for $V = 5\gamma$ and $\gamma = 0.1$: (**a**) variational product state ansatz used previously[11]; (**b**) tensor network method with $D = 2$; (**c**) $D = 3$; (**d**) $D = 4$; (**e**) $D = 5$. We see no antiferromagnetic phase in this region for $D = 6, 7, 8$ and 9. Numerically, we see that the population difference drops down to $\sim 10^{-9}$ as soon as the antiferromagnetic disappears, whereas it is $\sim 10^{-1}$ when we observe it

simulations we have also found this antiferromagnetic region up to $D = 5$, Fig. 3 for $V = 5\gamma$, $\gamma = 0.1$, where for comparison we also show earlier data for the variational ansatz[11] with product states (the correlated ansatz produced the a decrease in AF ordering upon including correlations, which is consistent with the disappearance of the AF phase for large bond dimensions). Quite surprisingly, however, we find no AF phase for $D = 6, 7, 8$, and 9 around this region. The AF phase thus disappears for large bond dimension and for these values of the parameters. Notice that, however, this does not rule out the possibility of an AF phase appearing at some other parameter region.

In addition, we have simulated a dissipative spin 1/2 *XYZ* model on an infinite 2D square lattice, with Hamiltonian $H = \sum_{\langle i,j \rangle} (J_x \sigma_x^{[i]} \sigma_x^{[j]} + J_y \sigma_y^{[i]} \sigma_y^{[j]} + J_z \sigma_z^{[i]} \sigma_z^{[j]})$, and the same jump operators $L_\mu = \sqrt{\gamma} \sigma^{[\mu]}$. This model has been analyzed recently by cluster mean-field and corner space renormalization methods[10,13]. In particular, a possible re-entrance of the ferromagnetic phase at large coupling has been discussed[10]. In our simulations at large bond dimension we found no signal of such an effect, Fig. 4 for results in the regime $J_x = 0.5$, $J_z = 1$, and $D = 4$. Larger bond dimensions did not change this, in agreement with earlier asymptotic results[10].

## Discussion

Here we presented a simple TN method to approximate steady states for 2D quantum lattice systems of infinite size. Our approach relies on the hypothesis that when the dissipative fixed-point attractor is strong, then it drives the simulation to a good approximation of the steady state. We benchmarked our method with dissipative Ising and *XYZ* models. Future applications include the engineering of topologically ordered states by dissipation in 2D quantum lattice systems. It could also be applied to finite-temperature states, provided that a microscopic model for the coupling to the heath bath is included. Finally, it would be interesting to understand these results in the context of area-laws for rapidly mixing dissipative quantum systems[62,63].

## Methods

**Tensor network methods.** We used several tensor network methods in this paper. Summarizing, we used PEPOs to represent mixed states, simple update for the real-time evolution, and corner transfer matrices to compute local observables in the thermodynamic limit. We also computed the operator-entanglement entropy using such methods, and by additionally simplifying the calculation of the eigenvalues of the reduced density matrix of a block using the tensors obtained from the simple update. A detailed explanation can be found in Supplementary Notes 1–4.

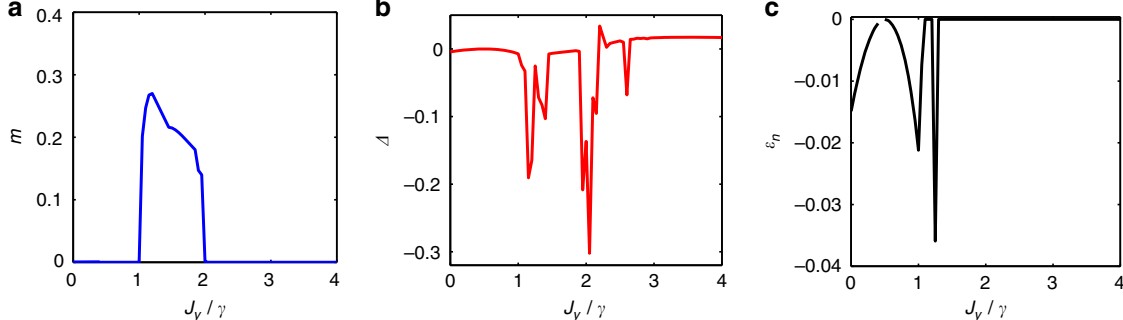

**Fig. 4** Ferromagnetic order parameter and error measures. This is computed for the *XYZ* model, for $J_x = 0.5$, $J_z = 1$ and $D = 4$, with the order parameter as an average of $|M_x| = |\langle \sigma_x \rangle|$ over the two sites $a$ and $b$ in our 2D PEPO construction of $\rho$, i.e., $m \equiv (|M_x^a| + |M_x^b|)/2$. In **a**, we observe no re-entrance of the ferromagnetic order $m$ at large values of $J_y/\gamma$. In **b**, we show $\Delta \equiv {}_\# \langle \rho_s | \mathcal{L}_\# | \rho_s \rangle_\#$, and in **c** we show $\varepsilon_n \equiv \sum_{i | \nu_i < 0} \nu_i(\rho_n)$ for $n = 4$ contiguous spins in a $2 \times 2$ plaquette. Larger errors as quantified by $\Delta$ and $\varepsilon_n$ appear around the phase transitions. Larger bond dimensions did not change the conclusion

**Code availability**. All numerical codes in this paper are available upon request to the authors.

**Data availability**. All relevant data in this paper are available from the authors.

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

## Acknowledgements

A.K. and R.O. acknowledge JGU, DFG GZ OR 381/1-1, DFG GZ OR 381/3-1, and discussions with I. McCulloch, A. Gangat, Y.-Jer Kao, M. Rizzi, D. Porras, J. Eisert, J. J. García-Ripoll, and C. Ciuti. H.W. acknowledges the Volkswagen Foundation, DFG SFB 1227 (DQ-mat) and SPP 1929 (GiRyd).

## Author contributions

All authors contributed to all aspects of this work.

## Additional information

**Competing interests:** The authors declare no competing financial interests.

