## [Peer Review File · Nature Communications]

Reviewers' comments:

Reviewer #1 (Remarks to the Author):

The paper presents a numerical evaluation of local expectation values of numerically obtained fixedpoint solutions of 2d quantum dissipative models. In particular, the models are Ising model as well as spin 1/2 XYZ model. The numerical technique applied is that of tensor network states. In particular, the iPEPS ansatz is used on top of a local purification ansatz. Namely, the original 2D lattice structure is assumed for a purification of the density matrix which is then described by an iPEPS. By doing so, the authors provide a new perspective on recent controversies associated to interacting Rydberg atoms. While there is not much elaboration in the form of theoretical justification for this approach, it is acceptable to take a "just do it" mentality to solving pressing problems in many-body quantum physics and the authors do so through their numerical evaluation. In this respect, the work is a natural generalization of Refs. [15, 16] where a similar approach is taken for 1D systems. The current work is novel in that it considers inherently 2D models and uses iPEPS to address this task. The authors are among the few research groups possessing a running iPEPS code which is much more involved than MPS code (DMRG).

The current article applies modern state of the art tensor network techniques to active research questions. I find that the article constitutes valuable work. However, there are several points to which I would like to draw the author's attention.

- I don't know if this was a typo or intended, but the abstract states

"We prove the validity ..."

There is no mathematical proof being provided so I suggest replacing "prove" by "probe", "test". As it currently stands, the sentence is overstating the degree of rigor with which the numerical approach is verified.

-In the introduction, the language used by the authors ("the") suggests that there may be a single steady state associated to a quantum dissipative system described by L . Indeed, one of the conclusions made in the article is the absence (with respect to numerical observation) of a bistable region. Admittedly, the literature is split with respect to whether the dissipative quantum Ising model admits a bistable region. The ansatz employed by the authors is based on finite number of parameters and the lack of ergodicity through bi(multi)stability is a thermodynamic phenomenon. It would thus, be appropriate for the work to include evidence that the current ansatz used is capable of diagnosing genuinely bistable phases. A similar use of "the" for steady state occurs near the end of the paper.

- The authors state that to check that the state obtained is a good approximation to a steady state, they calculate.

$$\langle \rho_s | L | \rho \rangle \approx 0$$

However, to my understanding, this is a necessary yet not sufficient condition. It only checks that the derivative of the state is perpendicular to the state itself, which should always be the case for a purity preserving evolution. The relevant test to perform would be

$$\langle \rho_s | L^\dagger | \rho \rangle = \langle \rho | L | \rho \rangle \approx 0$$

- In page two, the authors consider performing numerical evolution of the Liouville generator using trotter steps of 0.1 to 0.01. However, this comes before any unit is introduced for any of the terms involved in the dissipative evolution.

- The authors state that a 1st order phase transition is believed to exist for the dissipative Ising model. It would be adequate to indicate which references in the literature support this.

- The definition of purity for the reduced density matrix is missing a square, at the end of page 3.
- The fact that the ferromagnetic phase changes so significantly with bond dimension and disappears is somewhat mysterious.
- There are several typos, language and stylistic

"Ee see no AF phase ..." -> "We see no AF phase ..."

"... the study ... has boosted recently" -> "... has recently received much attention."

- It would be valuable to explain how negativity in the eigenvalue spectrum of the density matrix represented can arise and why no alternative options to mitigate this were considered.

It is my opinion that the article is presenting solid work and has good potential for publication. However, it is also my impression that it has been rushed and requires further curating to realize its potential.

Reviewer #2 (Remarks to the Author):

The authors propose a numerical method based on a tensor network (TN) scheme to study the steady states for 2D open quantum systems on a lattice in the thermodynamic limit. This is the first implementation of the calculation of steady states in 2D for quantum lattice systems with TNs. Understanding the effects of dissipation in quantum many-body systems is the focus of intensive research efforts due to possible applications in many disciplines. This manuscript should be of interests for researchers in the field of dissipative quantum manybody systems, and people working in TN algorithms.

The major claim of the manuscript is that one can directly apply known techniques for iPEPS to study the steady state, assuming the dissipation is strong enough to drive the evolution into a steady state that can be approximated by a TN, even when the entanglement is beyond the support of the TN. This is a very strong claim; however, the authors do not show this claim is generally true with a rigorous theoretical proof, nor numerically demonstrate it clearly for the quantum Ising model they study. For example, it is not demonstrated why the proposed method can by-pass a highly entangled intermediate state. Although the proposal is clearly new, they do not provide enough evidences to support their claim. With this, I could not recommend the publication of this manuscript in the present form.

Here I list a few comments that might help the authors improve their manuscript.

(1) In the introduction, the authors wrote "As time flows, the system dissipates, until reaching in many cases a steady, or "dark" state, which is the right eigenvector of L with zero eigenvalue." This sentence can be very confusing for non-specialists as the vectorized form of the density matrix is not introduced later in the text, and it is not possible to understand without prior knowledge of dissipative quantum systems.

(2) PEPO should be the acronym for Projected Entangled-Pair Operator.

(3) For the antiferromagnetic region, they compare their results with the data from Ref. [6] using variational ansatz with product states. It is not clear why the product state ansatz is chosen as in Fig. (2), the correlated state ansatz clearly gives a better result.

(4) In the conclusion, they claim that "Our approach relies on the hypothesis that, for some

systems, the dissipative fixed-point attractor is so strong that it drives the simulation to a good approximation of the steady state even when the intermediate time-evolved states have too much entanglement for the TN representation." It is not clear why this is generally true. The iPEPS steps, as far as I understand, still relies on a real-time evolution of PEPO, which potentially can go to a highly entangled transient state before a weakly entangled steady state is reached. By turning up the dissipation, it is possible that the system will not evolve into a highly entangled state, but for a transient state whose entanglement is beyond the support of the PEPO, I fail to see it should work. It is possible that for the parameters chosen by the authors, the system stays weakly entangled along the evolution trajectory. They should either explicitly show that the transient entanglement got very large when they applied their method, or give some rigorous theoretical argument that their method would work in such a situation.

Reviewer #3 (Remarks to the Author):

This manuscript introduces a novel algorithm allowing to study the steady state of driven-dissipative quantum systems on two dimensional lattices. The general framework is based on the manipulation of tensor networks; the method presented here is a simple reformulation of the infinite-PEPS algorithm applied to the time evolution of mixed states, following a Liouvillian master equation in the Lindblad form.

The study of open quantum many-body systems is currently a hot topic in the context of quantum statistical mechanics. Unfortunately, if we neglect the one-dimensional case, the available numerical methods that are able to go beyond a simple mean-field analysis are scarce, and are all affected by severe limitations (either in the simulatable system size, or in the accuracy of the results, especially when close to the critical behaviour).

The approach presented here might represent a significant step forward along this direction, thus being potentially relevant in order to guarantee the publication in Nature Communications. The results on two paradigmatic spin models are also important as an unbiased benchmarks for the existing algorithms, such as a correlated variational ansatz, the cluster mean-field, and the corner-space renormalization approach.

There are however several points in the paper that appear to me quite obscure, and need to be clarified.

The first one concerns the way in which the so called "projected entangled state operator" (why is it called PEPO and not PESO?) is manipulated, in order to perform the imaginary time evolution. It is quite clear and well explained how to perform the vectorization procedure, when going from pure states to mixed states. However the following section on "Computing 2d steady states" is basically just stating the main idea of the algorithm (which is indeed simple), and then referring to a number of technical papers. No explanation on how to apply the "iPEPS machinery" in this context is given. For example, it would be very useful to understand a little more how the "simple update scheme" for the time evolution works, or even how the tensor network structure is actually efficiently contracted for the calculation of observables (the "corner transfer matrices"). As far as I understand, this is indeed a very non trivial issue in any PEPS manipulation.

Besides that, it is not clear to me whether one can safely reach a not much entangled steady state (due to dissipation), even when starting from a random initial state, without getting trapped into some highly entangled state. The time evolution implemented here is some sort of quench, where the intermediate steps will be poorly described by the PEPO. Can one be sure that, when escaping out of the small-entanglement region during the time evolution, one could then eventually be restored into a "simulatable" low-entanglement state at long times (coinciding with the correct Liouvillian steady state)?

Concerning the results, it would be useful to have an idea of the times that are needed to reach the convergence. How many Trotter steps have been applied? Does this number crucially depend on the distance from some "critical behaviour" in the system, where the Liouvillian gap closes?

In the simulations of the 2D Ising model, is there a particular reason why one does not observe a monotonic convergence of the density as a function of the field, for finite bond-link dimension toward the asymptotic value?

The results on the XYZ Heisenberg model seem to agree with those presented in Ref.[10], since in both cases a reentrance of the paramagnetic phase at large coupling strength is observed.

It would be probably useful to comment on a similar approach that has been recently adopted in order to study 2D thermal states: [Czarnik et al., Phys. Rev. B 92, 035152 (2015); Phys. Rev. B 94, 235142 (2016)].

In conclusion, for the reasons mentioned above, I am generally inclined toward a positive judgment on the publication of this manuscript in Nature Communications. According to me, it is however important that the authors address all the points raised above, before recommending the publication.

We are glad that the overall assessment of the referees is quite positive with Referee 1 concluding that our work is “novel” and “valuable” and a “good potential for publication”. Referee 3 also concluded that the present work might represent a “significant step forward” along the direction of study of open quantum many-body systems and therefore “potentially relevant in order to guarantee the publication in Nature Communications”. Referee 2 is of the opinion that although the proposal is clearly new, there is not enough evidence to support the claim and hence cannot recommend publication in its present form. After taking into account the suggestions and comments by the referees, we have revised our manuscript accordingly including more evidences to support our claims. Below we also provide a response to the recommendations and criticisms by the referees, together with a list of changes. The changes in the new version are highlighted in red. With these modifications and further evidences, we hope that our manuscript will be further considered for publication in Nature Communications.

Referee 1

We thank the referee for his/her positive assessment on our work. We also appreciate the referee for the very useful comments and suggestions for improving our manuscript. Below we address individually the comments by the referee:

Referee said: “The current article applies modern state of the art tensor network techniques to active research questions. I find that the article constitutes valuable work. However, there are several points to which I would like to draw the author's attention.”

Reply: we would like to thank the referee for his/her valuable comment about our work.

Referee said: “I don't know if this was a typo or intended, but the abstract states “We prove the validity ...” There is no mathematical proof being provided so I suggest replacing “prove” by “probe”, “test”. As it currently stands, the sentence is overstating the degree of rigor with which the numerical approach is verified.”

Reply: we agree that there was no mathematical proof provided in the manuscript. We have therefore modified our sentence accordingly.

Referee said: “In the introduction, the language used by the authors (“the”) suggests that there may be a single steady state associated to a quantum dissipative system described by \mathcal{L} . Indeed, one of the conclusions made in the article is the absence (with respect to numerical observation) of a bistable region. Admittedly, the literature is split with respect to whether the dissipative quantum Ising model admits a bistable region. The ansatz employed by the authors is based on finite number of parameters and the lack of ergodicity through bi(multi)stability is a thermodynamic phenomenon. It would thus, be appropriate for the work to include evidence that the current ansatz used is capable of diagnosing genuinely bistable phases. A similar use of “the” for steady state occurs near the end of the paper.”

Reply: in the revised version we have included a detailed analysis of the bistable region in the new Fig.(2(e)). We find, in fact, a small bistable region for small bond dimension $D=1,2$, with the results for $D=1$ in agreement with mean field calculations. However this bistable region disappears with large bond dimension $D > 2$. We therefore conclude that there is no bistability for large bond dimension, and that it is an effect observed for small bond dimensions only.

Referee said: "The authors state that to check that the state obtained is a good approximation to a steady state, they calculate $\langle \rho_s | L | \rho_s \rangle \approx 0$. However, to my understanding, this is a necessary yet not sufficient condition. It only checks that the derivative of the state is perpendicular to the state itself, which should always be the case for a purity preserving evolution. The relevant test to perform would be $\langle \rho_s | L^\dagger | \rho_s \rangle = \langle L | \rho_s \rangle \approx 0$ "

Reply: we agree with the referee, but however, we also wish to point out that the calculation of $\langle \rho_s | L^\dagger | \rho_s \rangle$ is computationally more costly, very specially in 2d. This is the main reason why we focus on the calculation of $\langle \rho_s | L | \rho_s \rangle$ in our study. In the revised version we added the following comment about this: "Moreover, it should also be possible to check directly $\langle \rho_s | \mathcal{L}^\dagger | \rho_s \rangle$, but this is computationally more costly and does not change the conclusions of our observations."

Referee said: "In page two, the authors consider performing numerical evolution of the Liouville generator using trotter steps of 0.1 to 0.01. However, this comes before any unit is introduced for any of the terms involved in the dissipative evolution."

Reply: this is correct, and is only justified from experience. In a nutshell, we see that this choice works well for our purposes. We added the following comment about this in the paper: "The choice of Trotter steps actually depends on the time scales of the particular problem at hand. For the models considered here, we saw empirically that this choice was a good one."

Referee said: "The authors state that a 1st order phase transition is believed to exist for the dissipative Ising model. It would be adequate to indicate which references in the literature support this."

Reply: in the revised version of the paper we made more explicit the references pointing towards a first-order phase transition.

Referee said: "The definition of purity for the reduced density matrix is missing a square, at the end of page 3."

Reply: thank you, that was a typo, and we have corrected it in the revised version.

Referee said: "The fact that the ferromagnetic phase changes so significantly with bond dimension and disappears is somewhat mysterious."

Reply: we agree that it is somehow surprising, but indeed this is what we find. It's a discontinuous behavior with the bond dimension D .

Referee said: There are several typos, language and stylistic "Ee see no AF phase ..." -> "We see no AF phase ..." "... the study ... has boosted recently" -> "... has recently received much attention."

Reply: we corrected the typos in the revised version.

Referee said: "It would be valuable to explain how negativity in the eigenvalue spectrum of the density matrix represented can arise and why no alternative options to mitigate this were considered."

Reply: we included a comment about this in the revised version. In fact, this issue was already addressed in page 2 and in footnote 44 of the old version, but we stressed it even more in the revision. In particular, we added the following comment: "Moreover, in principle one could also develop a fully-positive algorithm for ρ_n \cite{werner}, but at the expense of accuracy and efficiency \cite{Gemma}"

Referee said: "It is my opinion that the article is presenting solid work and has good potential for publication. However, it is also my impression that it has been rushed and requires further curating to realize its potential."

Reply: we thank the referee for appreciating the value of our contribution, and hope that the revised version will address all his/her concerns.

Referee 2

We thank the referee for his valuable comments and for appreciating our work. In what follows we provide a detailed answer to his/her comments:

Referee said: "The authors propose a numerical method based on a tensor network (TN) scheme to study the steady states for 2D open quantum systems on a lattice in the thermodynamic limit. This is the first implementation of the calculation of steady states in 2D for quantum lattice systems with TNs. Understanding the effects of dissipation in quantum many-body systems is the focus of intensive research efforts due to possible applications in many disciplines. This manuscript should be of interests for researchers in the field of dissipative quantum manybody systems, and people working in TN algorithms. "

Reply: we would like to acknowledge the referee for appreciating the value of our contribution.

Referee said: "The major claim of the manuscript is that one can directly apply known techniques for iPEPS to study the steady state, assuming the dissipation is strong enough to drive the evolution into a steady state that can be approximated by a TN, even when the entanglement is beyond the support of the TN. This is a very strong claim; however, the authors do not show this claim is generally true with a rigorous theoretical proof, nor numerically demonstrate it clearly for the quantum Ising model they study. For example, it is not demonstrated why the proposed method can by-pass a highly entangled intermediate state. Although the proposal is clearly new, they do not provide enough evidences to support their claim. With this, I could not recommend the publication of this manuscript in the present form."

Reply: in the revised version we added two new plots in Fig.2(e) and Fig.2(f). The first one is a detailed study of the bistable region as a function of the bond dimension (see our reply to referee 1). The second one is the operator-entanglement entropy throughout the algorithm for increasing dissipation strength. This plot clearly shows how this quantity, which is the relevant one in terms of assessing the performance of the method, goes down with increasing dissipation, in turn implying more accuracy for the method. Our plot is for bond dimension $D=2$, but we repeated the calculation for larger bond dimensions up to $D=6$ with equivalent conclusion. This result proves our main claim: In a strongly-dissipative regime, the large entanglement region is bypassed before entanglement explodes, therefore allowing for accurate results with our method. Moreover, in the revised version we provide several pages of supplementary information on technical details of the algorithm, including the definition and procedure to compute the operator-entanglement entropy. We hope that our new results will convince the referee about our claims.

Referee said: “In the introduction, the authors wrote “As time flows, the system dissipates, until reaching in many cases a steady, or “dark” state, which is the right eigenvector of L with zero eigenvalue.” This sentence can be very confusing for non-specialists as the vectorized form of the density matrix is not introduced later in the text, and it is not possible to understand without prior knowledge of dissipative quantum systems.”

Reply: we modified this sentence in the revised version. Now it reads as follows: “As time flows, the system dissipates, until reaching in many cases a ρ_s , so that $\mathcal{L}[\rho_s] = 0$.”

Referee said: “PEPO should be the acronym for Projected Entangled-Pair Operator.”

Reply: thanks, that was a typo. We corrected it in the new version.

Referee said: “For the antiferromagnetic region, they compare their results with the data from Ref. [6] using variational ansatz with product states. It is not clear why the product state ansatz is chosen as in Fig. (2), the correlated state ansatz clearly gives a better result.”

Reply: we observe that with the correlated ansatz, the magnitude of the AF order parameter decreases, which is compatible with the disappearance of this phase for large bond dimension in our method. We added a comment about this in the paper: “The correlated ansatz produced a decrease in AF ordering upon including correlations, which is consistent with the disappearance of the AF phase for large bond dimensions.”

Referee said: “In the conclusion, they claim that “Our approach relies on the hypothesis that, for some systems, the dissipative fixed-point attractor is so strong that it drives the simulation to a good approximation of the steady state even when the intermediate time-evolved states have too much entanglement for the TN representation.” It is not clear why this is generally true. The iPEPS steps, as far as I understand, still relies on a real-time evolution of PEPO, which potentially can go to a highly entangled transient state before a weakly entangled steady state is reached. By turning up the dissipation, it is possible that the system will not evolve into a highly entangled state, but for a transient state whose entanglement is beyond the support of the PEPO, I fail to see it should work. It is possible that for the parameters chosen by the authors, the system stays weakly entangled along the evolution trajectory. They should either explicitly show that the transient entanglement got very large when they applied their method, or give some rigorous theoretical argument that their method would work in such a situation.”

Reply: we included the new Fig.2(f) in the revised version to address exactly this (for $D=2$, but we saw the same behavior for larger bond dimensions). In our simulations, as shown in the plot, we see that the transient states always have more operator-entanglement than the steady states, and that this operator-entanglement gets very much reduced when increasing the dissipation, in turn supporting our claim. As we explain in the supplementary material, we considered the operator entanglement because it is the relevant quantity to quantify the required size of the bond dimension in our algorithm, and therefore assess the accuracy of the method.

Referee 3

We thank the referee for his valuable comments and for appreciating the significance of our work. In what follows we reply to his/her comments.

Referee said: "The approach presented here might represent a significant step forward along this direction, thus being potentially relevant in order to guarantee the publication in Nature Communications. The results on two paradigmatic spin models are also important as an unbiased benchmarks for the existing algorithms, such as a correlated variational ansatz, the cluster mean-field, and the corner-space renormalization approach."

Reply: we would like to thank the referee for appreciating the potential and significance of our method.

Referee said: "The first one concerns the way in which the so called "projected entangled state operator" (why is it called PEPO and not PESO?) is manipulated, in order to perform the imaginary time evolution."

Reply: that was a typo and we corrected it in the revised version.

Referee said: "It is quite clear and well explained how to perform the vectorization procedure, when going from pure states to mixed states. However the following section on "Computing 2d steady states" is basically just stating the main idea of the algorithm (which is indeed simple), and then referring to a number of technical papers. No explanation on how to apply the "iPEPS machinery" in this context is given. For example, it would be very useful to understand a little more how the "simple update scheme" for the time evolution works, or even how the tensor network structure is actually efficiently contracted for the calculation of observables (the "corner transfer matrices"). As far as I understand, this is indeed a very non trivial issue in any PEPS manipulation."

Reply: we initially omitted this information because of space constraints. However, in the revised version we added supplementary material with all the technical details of the algorithm, as well as on how to compute the relevant quantities that we discuss in the paper.

Referee said: "Besides that, it is not clear to me whether one can safely reach a not much entangled steady state (due to dissipation), even when starting from a random initial state, without getting trapped into some highly entangled state. The time evolution implemented here is some sort of quench, where the intermediate steps will be poorly described by the PEPO. Can one be sure that, when escaping out of the small-entanglement region during the time evolution, one could then eventually be restored into a "simulatable" low-entangled state at long times (coinciding with the correct Liouvillian steady state)?"

Reply: yes, indeed this is what we see. To show this more clearly we added the new Fig.2(f) in our paper. It shows that the operator-entanglement in the transient states decreases with increasing dissipation strength, and that the operator-entanglement in the final steady states is not too large. As explained in the supplementary material (see also the replies to referees 1 and 2 above), this is the relevant measure of correlations in order to assess the performance of our method.

Referee said: "Concerning the results, it would be useful to have an idea of the times that are needed to reach the convergence. How many Trotter steps have been applied? Does this number crucially depend on the distance from some "critical behaviour" in the system, where the Liouvillian gap closes?"

Reply: the new Fig.2(f) in the revised version gives an idea about this. In practice we see that the convergence is quite fast and that the larger the gap of the Liouvillian, the faster the convergence. We added the following comment about this in the paper: "The convergence in the number of steps depends on the gap of the Liouvillian: the closer to a gapless point, the slower the convergence. Empirically we observed that this convergence was quite fast in the gapped phases of the models that we studied."

Referee said: "In the simulations of the 2D Ising model, is there a particular reason why one does not observe a monotonic convergence of the density as a function of the field, for finite bond-link dimension toward the asymptotic value?"

Reply: this has to do with the simple update scheme used in our calculations. As is well known, this update is extremely efficient, but it may also lead to such behavior, even though it produces good results in the large-D limit. More accurate results could be obtained with more refined and less efficient updates, such as the full and fast full updates, and this would certainly provide a smoother convergence. We did not explore this numerically (hence the title "a simple...") and consider it for future works. A comment about this was already included in the caption of Fig.2 in the old version of the paper: "Overall, the convergence can be further improved by using more accurate update schemes"

Referee said: "The results on the XYZ Heisenberg model seem to agree with those presented in Ref.[10], since in both cases a reentrance of the paramagnetic phase at large coupling strength is observed."

Reply: this is correct. We modified the discussion accordingly.

Referee said: "It would be probably useful to comment on a similar approach that has been recently adopted in order to study 2D thermal states: [Czarnik et al., Phys. Rev. B 92, 035152 (2015); Phys. Rev. B 94, 235142 (2016)]."

Reply: we added a comment about this in the paper as well as the references. Notice however that in our case we focus on steady states, which do not need to be thermal.

Referee said: "In conclusion, for the reasons mentioned above, I am generally inclined toward a positive judgment on the publication of this manuscript in Nature Communications."

Reply: we thank the referee for appreciating the value of our work.

List of changes (highlighted in red in the new version)

- 1) We added two new plots: Fig.2(e) studying the bistable region, and Fig.2(f) with the evolution of the operator-entanglement entropy throughout the algorithm.
- 2) We added several pages of supplementary material in pdf, with all the technical details about PEPOs, the simple update, calculation of local observables with corner transfer matrices, and the definition as well as different ways to compute the operator-entanglement entropy.
- 3) We modified several small typos, including all the typos observed by the referees.
- 4) We added several small modifications of the text taking into account each one of the comments made by the referees, as specified in the individual replies above.

Reviewers' comments:

Reviewer #2 (Remarks to the Author):

In the revised manuscript, the authors included supplementary materials explaining the technical details of various tensor network computation mentioned in the main text and added Fig. 2(e) and (f) clarified some of the points raised by me and the other referees.

In general, the major concerns I raised were addressed by the authors. In particular, from the operator entanglement entropy plot Fig. 2(f), it is clear that the method works only in the limit of strong dissipation such that the entanglement of the transient state can be supported by the PEPO, not as claimed in the conclusion "even when the intermediate time-evolved states have too much entanglement for the TN representation." Following Eq. (7) in the Supplementary Materials, the entanglement shown in Fig. 2(f) never exceeds the entanglement support of the PEPO $4 \times 2 \times \ln(2) = 5.545$.

The non-monotonic bond dimension dependence of the transition in Fig. 2(a) remains mysterious and unexplained.

To summarize, as stated by the other referees, I am also inclined toward a positive judgment on the publication of this manuscript in Nature Communications. However, I would hope the authors to tone down their claim on the simulability using TN as the evidence suggests otherwise.

Reviewer #3 (Remarks to the Author):

After reading the reply letter and the changes made by the authors, I think that the paper has been improved and I am now satisfied with it.

The added supplementary material can be helpful to anyone who is interested in writing a functioning algorithm, which can also reproduce the results contained here.

In conclusion, I can now recommend the publication in Nature Communications.

Referee 2

We thank the referee for his positive judgement of our work, as well as for his comments. In the revised version we addressed the main two concerns mentioned in his/her last report. First, we softened our claims concerning the regions where our algorithm is applicable, making it very explicit that it works only in the strong dissipation regime, when entanglement is small enough to be supported by the PEPO. Second, we made it clear that there is a non-monotonic convergence with D in Fig.2(a), probably due to the effect of the different approximations of the algorithm in the transition region, and which remains to be fully understood.

We implemented several modifications accordingly, as specified in the list of changes.

List of changes (highlighted in red in the new version)

- 1) We removed the sentence “even when the intermediate time-evolved states have too much entanglement for the TN representation” from the conclusions.
- 2) We removed sentence “In fact, even if there is too much entanglement for the TN, the dissipation may still drive the evolution towards a good approximation of the correct steady state.” when explaining the method.
- 3) We added in the abstract the following sentence: “and is based on the intuition that strong dissipation kills quantum entanglement before it gets too large to handle.”

4) We modified the following sentence in the results section: “The stronger the dissipation, the weaker the operator entropy (which never exceeds the support of the PEPO), and therefore the better the performance of the algorithm, as claimed.”

5) We added the following sentence in the results section: “We also observe a non-monotonic convergence in $\$D\$, which may be due to a stronger effect of the approximations in the transition region, and which remains to be fully understood.”$